# Quercetin Inhibits the Proliferation and Aflatoxins Biosynthesis of *Aspergillus flavus*

**DOI:** 10.3390/toxins11030154

**Published:** 2019-03-09

**Authors:** Xiu-Mei Li, Zhong-Yuan Li, Ya-Dong Wang, Jin-Quan Wang, Pei-Long Yang

**Affiliations:** 1Key Laboratory of Feed Biotechnology, Ministry of Agriculture and Rural Affairs, Feed Research Institute, Chinese Academy of Agricultural Sciences, Beijing 100081, China; wangjinquan@caas.cn; 2National Engineering Research Center of Biological Feed, Beijing 100081, China; 3State Key Laboratory of Food Nutrition and Safety, Key Laboratory of Industrial Fermentation Microbiology of Ministry of Education & Tianjin Key Laboratory of Industrial Microbiology, College of Biotechnology, Tianjin University of Science and Technology, Tianjin 300457, China; lizhongyuan@tust.edu.cn (Z.-Y.L.); wangyadong1209@163.com (Y.-D.W.)

**Keywords:** quercetin, *Aspergillus flavus*, transcriptome, RNA-seq, aflatoxin

## Abstract

In this work of quercetin’s anti-proliferation action on *A. flavus*, we revealed that quercetin can effectively hamper the proliferation of *A. flavus* in dose-effect and time-effect relationships. We tested whether quercetin induced apoptosis in *A. flavus* via various detection methods, such as phosphatidylserine externalization and Hoechst 33342 staining. The results showed that quercetin had no effect on phosphatidylserine externalization and cell nucleus in *A. flavus*. Simultaneously, quercetin reduced the levels of reactive oxygen species (ROS). For a better understanding of the molecular mechanism of the *A. flavus* response to quercetin, the RNA-Seq was used to explore the transcriptomic profiles of *A. flavus*. According to transcriptome sequencing data, quercetin inhibits the proliferation and aflatoxin biosynthesis by regulating the expression of development-related genes and aflatoxin production-related genes. These results will provide some theoretical basis for quercetin as an anti-mildew agent resource.

## 1. Introduction

*Aspergillus flavus* is a saprophytic filamentous fungus that produces aflatoxins (AF), which are mutagenic, teratogenic and carcinogenic toxins for humans and animals [1,2,3,4]. Currently, there is a large amount of natural products, synthetic compounds, and extracts from diverse organisms for inhibitors of *A. flavus* growth, and aflatoxin biosynthesis that were investigated for application in food and feed preservation due to their low impact on the environment and human health [5,6,7,8,9,10].

The addition of anti-mildew agent is one of the important measures to prevent mildew pollution. Natural anti-mildew agent is a more ideal choice. Quercetin (3,3′,4′,5,7-pentahydroxy-flavone) is a natural resource found in many plants, fruits and vegetables [11]. Due to its anti-oxidant [12], anti-inflammatory [13], anti-cancer [5], antiviral, antibacterial [11], and anti-proliferative activity [5,11] and so on, it has been chemically synthesized and commercially sold. Previous studies revealed that quercetin could inhibit the proliferation and AF biosynthesis of *A. flavus* [6]. However, the molecular mechanisms are still not well-clarified. In this work, we hope to reveal the potential mechanism by which quercetin inhibits the proliferation and AF biosynthesis of *A. flavus*. It provides a theoretical basis for quercetin as an anti-mildew agent.

## 2. Results

### 2.1. Quercetin Inhibited the Proliferation of *A. flavus*

In our works, as shown in Figure 1A–C, quercetin caused a markedly dose-effect and time-effect reduction in *A. flavus* cells viability, with the MIC value at 505 μg/mL. Next, we attempted to estimate the minimum bactericidal concentration (MBC) value; under the concentration of MIC (505 μg/mL), the single colony was not found in the potato dextrose agar (PDA) plates. It can be seen from this that when the concentration of MIC was 505 μg/mL, the spore survival rate was zero. Therefore, the MBC value was the same as the MIC (Figure 1D,E). Therefore, we conclude that quercetin might inhibit the proliferation of *A. f lavus.*

### 2.2. Morphological Changes of *A. flavus*

Spore (10^7^ CFU/mL) were treated with quercetin (200 μg/mL). After 24 h, *A. flavus* cells were harvested. The morphological changes of *A. flavus* were observed with the aid of a microscope with a 100-fold oil mirror. The result is shown in Figure 2. Compared with the control group, the mycelia of *A. flavus* were significantly degraded in the quercetin treated group.

### 2.3. Cell Apoptosis

We used annexin-V-FITC/propidium iodide (PI) double staining to differentiate intact cells from non-apoptotic cells (annexin-V negative and PI negative), early apoptotic cells (annexin-V positive and PI negative), late apoptotic cells (annexin-V positive and PI positive), and dead (necrotic) cells (PI positive) and to examine apoptosis more deeply [14]. As shown in Figure 3A, only the quercetin-treated group produced dead (necrotic) cells. The observations suggested that *A. flavus* cells have died via necrosis but not through the apoptotic pathway. In addition, compared with the control group, quercetin did not cause changes in the nuclear integrity of *A. flavus* (Figure 3B). Generation of ROS happens at the onset of apoptosis [5,15]. However, in our work, quercetin did not cause reactive oxygen species to rise, but caused reactive oxygen species to decrease (Figure 3C). This further indicates that quercetin does not induce the death of *A. flavus* through apoptotic pathway. In conclusion, these results demonstrated that quercetin does not induce apoptosis in *A. flavus.*

### 2.4. RNA-Seq Data

The transcriptome of *A. flavus* was put together from scratch with paired-end raw reads brought forth by the Illumina HiSeq2500 instrument. After redundancy and short reads had been weeded out, the clean reads in the QT group and CK group were 50561156 and 51441686, respectively (Appendix A). The Illumina guidelines were used to sequence data for every sample found to have Q30 as its quality score. The GC counts for the QT group and the CK group were 52.39% and 52.33%, respectively (Appendix A). Also, 45577031 (90.14%) and 46843066 (89.88%) clean reads that we got from the two groups effectively matched the value for the *A. flavus* genome. 89.42% of the reads were individually mapped to the genome for the QT group and 89.68% for CK group (Appendix A) according to the statistics. Moreover, 0.46% and 0.46% of the reads were multiply mapped to the genome for the CK group and the QT group, respectively (Appendix A). These results showed that the sequencing quality was suitable for the unigenes of subsequent annotation analysis.

### 2.5. Identification and Functional Annotation

From the FPKM (Reads Per Kilobase of exon model per Million mapped reads) values, we identified 665 differentially expressed genes (log2[fold change] = log2[QT/CK] > 1, Probability > 0.8) between the QT and CK groups. Of these, 340 genes up-regulated and 325 genes down-regulated following exposure to quercetin (Appendix A). We carried out a GO functional enrichment analysis of these differently expressed genes. The results demonstrated that these genes played a role in structural constituent of ribosome, structural molecule activity, electron carrier activity, rRNA binding, cis-trans isomerase activity, translation, cellular protein metabolic process, protein metabolic process, cellular biosynthetic process, biosynthetic process, organic substance biosynthetic process, cellular metabolic process, cellular macromolecule biosynthetic process, gene expression, macromolecule biosynthetic process, organic substance metabolic process, primary metabolic process, cellular macromolecule metabolic process, cellular process, organonitrogen compound biosynthetic process, purine nucleoside triphosphate biosynthetic process, purine ribonucleoside triphosphate biosynthetic process, ribosome, ribonucleoprotein complex, non-membrane-bounded organelle, intracellular non-membrane-bounded organelle, cytoplasmic part, macromolecular complex, cytoplasm, cell, cell part, intracellular part, intracellular, organelle, intracellular organelle, ribosomal subunit, small ribosomal subunit, or proton-transporting ATP synthase complex (Appendix A). KEGG (Kyoto Encyclopedia of Genes and Genomes) metabolic pathway enrichment analysis shown that these genes were primarily involved in the ribosome, Oxidative phosphorylation, Huntington’s disease, Parkinson’s disease and Alzheimmer’s disease (Appendix A). Our analysis of KEGG metabolic pathway enrichment showed that these genes played a role in the ribosome, Huntington’s disease, Oxidative phosphorylation, Parkinson’s disease and Alzheimer’s disease (Appendix A).

### 2.6. Expression Analysis of Conidial Development- and *A. flavus* Growth-Related Genes in Response to Quercetin

To elucidate the effects of quercetin on the regulation of conidia and mycelia, based on the differentially expression genes (Appendix A) of *A. flavus* in the CK and QT groups, we found that some genes that played a role in conidial and mycelial development were down-regulated when quercetin was used (Table 1), including sexual development transcription factor NsdD (AFLA_020210), sexual development transcription factor SteA (AFLA_048650), G protein complex alpha subunit GpaB (AFLA_018540), APSES transcription factor StuA (AFLA_046990), conidiation-specific protein Con-10 (AFLA_083110) and conidiation-specific family protein (AFLA_044790).

Ribosomal proteins (RPs) are needed for all types of cells to grow and survive [16,17]. In our study, the analysis of KEGG metabolic pathway enrichment demonstrated that the ribosome was the most deregulated metabolic pathway in *A. flavus* that underwent treatment with quercetin, as shown in Table 2, 65 of differentially expressed genes were significant regulated, including 60 genes down-regulated (RPS6, RPS12, RPS13, RPS14, RPS15, RPS16, RPS17, RPS18, RPS19, RPS2, RPS20, RPS21, RPS22, RPS23, RPS24, RPS25, RPS26, RPS28, RPS4, RPS5, RPS8, RPS9, RPP0, RPP1, RPP2, RPL1, RPL11, RPL12, RPL13, RPL14, RPL15, RPL16, RPL18, RPL2, RPL20, RPL21, RPL23, RPL24, RPL25, RPL26, RPL27, RPL28, RPL3, RPL30, RPL31, RPL32, RPL33, RPL34, RPL35, RPL36, RPL37, RPL38, RPL42, RPL43, RPL5, RPL6, RPL7 or RPL8 ) and 5 genes up-regulated (RPL17, RPL19, RPL22, RPL32 or RPL9). Interestingly, all the 5 up-regulated genes were a protein component of the 60S large ribosomal subunit.

### 2.7. Expression Analysis of *A. flavus* AF Biosynthesis-Related Genes in Response to Quercetin

To elucidate the effects of quercetin on the regulation of aflatoxin biosynthesis, based on the analysis of differentially expressed genes data of *A. flavus* in the CK and QT groups (Appendix A), the transcription regulator gene *aflS* (AFLA_139340) was significantly down-regulated. Further, *aflS* gene was validated by real-time RT-PCR analysis. The data confirmed the significant down-regulated of gene *aflS* (Figure 4), which was consistent with transcriptome data (Appendix A).

## 3. Discussion

Quercetin is one of the natural flavonoids that play a crucial role in antibacterial activity [11]. Studies of flavonoid molecules of structure activity relationship have demonstrated that the oxygen atoms at position 4 in the C ring and the hydroxyl at positions 5 and 7 in the A ring constitute the primary group of antibacterial activity; next to them is the hydroxyl at position 3 in the C ring for antibacterial activity of such compounds. The hydroxyl at positions 3′ and 4′ in the B ring also shows some antibacterial activity [11]. Quercetin includes the oxygen atoms at position 4 in the C ring and the hydroxyl at position 5, 7, 3, 3′ and 4′. Previous studies have shown that quercetin can inhibit the proliferation of *A. flavus* [6]. However, in our work, we found that quercetin not only inhibited the growth of *A. flavus*, but also killed *A. flavus*, with a minimum inhibitory concentration of 505 μg/mL, and a minimum fungicidal concentration of 505 μg/mL (Figure 1). In addition, quercetin inhibits the growth of *A. flavus* in a dose-effect and time-effect relationship (Figure 1C).

Apoptosis is a kind of physiological programmed cell death and is different from necrosis [18]. One important mechanism referred to the function of antifungal drugs is the activation of the apoptotic pathway [7,19,20]. Antifungal agents trigger morphological features characteristic of apoptosis including PS externalization, nuclear condensation and ROS generation and so on, when they induce apoptosis on fungi [7,18,19,20]. However, in our works, the results of Annexin V-FITC/PI staining shown only the quercetin-treated group produced dead (necrotic) cells (Figure 3A). Subsequently, the morphological features characteristics of nuclear condensation was observed, we found that compared with the control group, quercetin did not cause changes in the nuclear integrity of *A. flavus* (Figure 3B). In addition, generation of ROS happens at the onset of apoptosis [5,15], in our work, quercetin did not cause reactive oxygen species to rise, but caused reactive oxygen species to decrease (Figure 3C). This result is contrary to the PS externalization morphological features characteristics of apoptosis. From the above, we concluded that quercetin might not induce apoptosis in *A. flavus.* How does quercetin inhibit the growth of *A. flavus*? We used transcriptome sequencing to reveal its possible mechanism. In our work, the mechanism by which quercetin inhibits *A. flavus* proliferation and aflatoxin biosynthesis was investigated adopting an RNA-seq analysis.

Based on our transcriptome data, we found that some genes that played a role in conidial and mycelial development were down-regulated when quercetin was used (Table 1), including sexual development transcription factor NsdD (AFLA_020210), sexual development transcription factor SteA (AFLA_048650), G protein complex alpha subunit GpaB (AFLA_018540), APSES transcription factor StuA (AFLA_046990), conidiation-specific protein Con-10 (AFLA_083110) and conidiation-specific family protein (AFLA_044790). When the development of *A. flavus* is inhibited, the sexual development transcription factor NsdD (AFLA_020210) [9] and SteA (AFLA_048650) were significantly down-regulated. Concurrently, transcriptions of conidia-specific genes, such as conidiation-specific family protein (AFLA_044790) and Con-10 (AFLA_083110) were significantly down-regulated [21]. The APSES transcription factor StuA that affects the orderly differentiation and spatial organization of cell types of the conidiospore [8] is encoded by transcription of the *stuA* gene (AFLA_046990), and the G protein complex alpha subunit GpaB (AFLA_018540) was significantly decreased. During aflatoxin biosynthesis, AflR is essential for expression of most of the genes in the aflatoxin genes cluster [6], which AflS (AFLA_139340) was reported to interact with activating AflR to give play to its regulatory effect [22]. As is known to all that fungal growth was closely related to biosynthesis of secondary metabolism [2,23]. APSES transcription factor StuA to be required for fungal conidial and mycelium growth [8,24]. Down-regulation of APSES transcription factor StuA inhibited the aflatoxin biosynthesis [24]. In our works, the transcription regulator genes *aflS* were significantly down-regulated (Appendix A and Figure 4). In addition, the redox state in the mycelia of *A. flavus* has been proved to be closely related to aflatoxin production [6]. Quercetin reduced the ROS level in the *A. flavus* (Figure 3C). So, quercetin may reduce the production of aflatoxin by lowering levels of ROS.

Ribosomal proteins (RPs) are needed for all types of cells to grow and survive [16,17]. In eukaryotic cells, the ribosome is made up of two subunits, a large subunit (60S) and a small subunit (40S) [16,25,26]. The small subunit (40S) is the t-RNA binding, decoding, and mRNA passage site [27,28]. The large subunit (60S) afforded a GTPase binding platform, polypeptide exit tunnel and peptidyl transfer [25,26]. In this study, the analysis of KEGG metabolic pathway enrichment demonstrated that the ribosome was the most deregulated metabolic pathway in *A. flavus* that underwent treatment with quercetin, including RPS6, RPS12, RPS13, RPS14, RPS15, RPS16, RPS17, RPS18, RPS19, RPS2, RPS20, RPS21, RPS22, RPS23, RPS24, RPS25, RPS26, RPS28, RPS4, RPS5, RPS8, RPS9, RPP0, RPP1, RPP2, RPL1, RPL11, RPL12, RPL13, RPL14, RPL15, RPL16, RPL17, RPL18, RPL19, RPL2, RPL20, RPL21, RPL22, RPL23, RPL24, RPL25, RPL26, RPL27, RPL28, RPL3, RPL30, RPL31, RPL32, RPL33, RPL34, RPL35, RPL36, RPL37, RPL38, RPL42, RPL43, RPL5, RPL6, RPL7, RPL8 or RPL9 that regulates ribosomal proteins (Table 2) without interfering with nucleolar integrity (Figure 3B). RPP0 binds permanently to the 60S subunit, and it is a necessary protein for ribosome function and structure because its exclusion would kill the cell [27]. RPP0 inhibits cell proliferation when it is down-expressed [26]. The 60S large ribosomal subunit has RPL23 as a protein component. Interestingly, RPL23 reportedly causes growth inhibition and has anti-tumor effects in gastric cancer, SKM-1 and K562 cells, when it is suppressed [28,29]. RPL27 and RPL30 have possess antimicrobial properties against *Streptococcus uberis*, *Streptococcus pyogenes* and *Enterococcus faecium* [30]. The 60S large ribosomal subunit has RPL17 as a protein component. Previous studies have shown that the over-expression of RPL17 inhibits cell growth and proliferation, while not affecting cell apoptosis [31]. Over-expression of RPL19 is implicated in lower prostate cancer survival. On the contrary, the role of RPL19 in promoting tumor formation was confirmed using transient and stable knockdown of RPL19 mRNA [32,33]. RPL22 is a protein component of the 60S large ribosomal subunit. Overexpression of RPL22/eL22 leads to the increase of p53, p21 and MDM2 protein levels, which RPL22/eL22 can suppress cancer cell proliferation and growth in a p53-dependent fashion [34]. Down-regulation of ribosomal protein L34 (RPL34) could hamper the multiplication of esophageal cancer cells [35,36]. Down-expression of RPL6 and RPS13 inhibit cell proliferation and cell cycle progression in gastric cancer cells [37,38]. Over-expression of RPL9 inhibits rabies virus replication [39]. The function of some ribosomal proteins is still unknown, which requires further study.

## 4. Materials and Methods

### 4.1. Reagents

The quercetin (purity > 98.0%) was bought from the National Institutes for Food and Drug Control (Beijing, China). Muse® Oxidative Stress Assay Kit was bought from Merckmillipore (Billerica, MA, USA). Hoechst 33342 and Annexin V-FITC Kit were bought from Beyotime (Shanghai, China).

### 4.2. Fungus Strain and Cultivation

*A. flavus* (CGMCC3.6434) was bought from the China General Microbiological Culture Collection Center (CGMCC, Beijing, China). The *A. flavus* was cultured at 28 °C in a potato dextrose agar (PDA) and preserved in a refrigerator at 4 °C.

### 4.3. Anti-Proliferative Activity

Colorimetric 3-(4,5-dimethylthiazol-2-yl)-2,5-diphenyltetra-azolium bromide (MTT) assay was applied to measure the proliferation of *A. flavus*. Spore (10^7^ CFU/mL) was inoculated into Sabouraud’s Glucose Broth Medium at 200 mL/well in 96-well microtiter plates. Two-fold consecutive dilutions of quercetin (0, 50, 100, 200, 400, 800 μg/mL) were made to wells inhabited by spore. After incubating for 24 h at 30 °C, each concentration was assayed in triplicate (*n* = 3). 24 h later, 10 μL of the MTT (5 mg/mL) reagent was put in each well and the plates were left to incubate at 30 °C for 4 h. Then, the reaction was ended by the addition of DMSO (100 μL), and the plate was agitated a little to redissolve the formed crystals. The absorbance of each well was assessed with a Multiskan Sky microplate reader (Thermo Scientific, Waltham, MA, USA). The results appeared as the inhibition ratio of cell multiplication calculated as [(A − B)/A] × 100% (A and B are the average numbers of viable cells of the control and samples, respectively).

In addition, the dose-effect and time-effect relationship experiments of quercetin on the proliferation of *A. flavus* were determined. Spore (10^7^ CFU/mL) was inoculated into Sabouraud’s Glucose Broth Medium at 200 μL/well in 96-well microtiter plates. Various concentrations (0, 50, 100, 200, 400, 800 μg/mL) of quercetin were put in wells inhabited by spore. The light absorption value at 600 nm was detected every two hours using Multiskan Sky microplate reader (Thermo Scientific, Waltham, MA, USA).

Minimum bactericidal concentration (MBC) refers to the least concentration of bactericide required to kill 99.9% of bacteria inoculums. Briefly, after the anti-proliferative activity assay, the *A. flavus* solution was sucked out of the 96-well plate, centrifuged at 8000 rpm for 5 min, washed with quercetin, and then suspended the *A. flavus* with 0.9% normal saline. The washed *A. flavus* was then coated onto PDA plates, and the single colony was cultured at 30 °C for counting. The results appeared as the inhibition ratio of cell proliferation calculated as [(A − B)/A] × 100% (A and B are the average numbers of viable cells of the control and samples, respectively).

### 4.4. Morphological Changes of *A. flavus*

Spores (10^7^ CFU/mL) were treated with quercetin (200 μg/mL). After 24 h, *A. flavus* cells were observed with a CX31 microscope with 100-fold oil mirror (Olympus, Tokyo, Japan).

### 4.5. Phosphatidylserine (PS) Externalization

Spores (10^7^ CFU/mL) were treated with quercetin (200 μg/mL). After 24 h, *A. flavus* cells were harvested and stained for 30 min with fluoresced in isothio-cyanate (FITC)-Annexin V and propidium iodide (PI) at room temperature in the dark, in line with the manufacturer’s recommendations (Beyotime, Shanghai, China). The cells were analyzed by Axio Vert A1 fluorescence microscope (Carl Zeiss, Jena, Germany).

### 4.6. Hochest 33342

Spore (10^7^ CFU/mL) were treated with quercetin (200 μg/mL). After 24 h, *A. flavus* cells were harvested and stained for 30 min with fluoresced Hoechst 33342 in the dark at 37 °C, according to the manufacturer’s recommendations (Beyotime, Shanghai, China). The cells were analyzed using Axio Vert A1 fluorescence microscope (Carl Zeiss, Jena, Germany).

### 4.7. Measurement of Reactive Oxygen Species

Spores (10^7^ CFU/mL) were treated with quercetin (200 μg/mL). After 24 h, *A. flavus* cells were harvested. The *A. flavus* cell walls were digested with 1.5% nailase (Solarbio, Beijing, China) and 1.5% Lyticase (Sigma, St Louis, MO, USA) and 1.5% cellulase (Onozuka, Tokyo, Japan) at 30 °C on a rotary shaker (80 rpm) for 3 h. They were washed twice in PBS and filtered through five layers of sterile lens paper to eliminate mycelial debris; then the protoplasts were obtained. According to the recommendations of the manufacturer, prepare cell samples in 1× assay buffer at 1 × 10^6^ CFU/mL, and then 10 μL of prepared cells were add to 190 μL of oxidative stress working solution. Incubate at 37 °C for 30 min. the cells were analyzed using Muse® Cell Analyzer (Merck, MA, USA).

### 4.8. cDNA Preparation and Illumina Sequencing

Construction of library and RNA-Seq were performed at Realbio Technology (Shanghai, China). Total RNA from quercetin-untreated (CK) and quercetin-treated (QT) groups was isolated with TRIzol Reagent (Invitrogen, Shanghai, China) following the recommendations of the manufacturer. The integrity and total concentration of RNA were assessed with a NanoDrop (Implen, Westlake Village, CA, USA), a Qubit® Fluorometer 2.0, and an Agilent 2100 Bioanalyzer (Agilent Technologies, Santa Clara, CA, USA) instruments. The mRNA was separated with the use of oligo (dT)-attached magnetic beads. The separated mRNA and the fragmentation buffer were mixed and cut into tiny fragments using divalent cations under high temperatures. The cDNA was synthesized with these cleaved RNA fragments as templates. Afterward, the short fragments and the adapters were connected. The fragments found suitable were picked as templates for the amplification of PCR. During the QC steps, Agilent 2100 Bioanaylzer (Agilent Technologies, Santa Clara, CA, USA) and ABI StepOnePlus Real-Time PCR System (Applied Biosystems, Waltham, MA, USA) were exploited for the qualification and quantification of the sample library. Lastly, the library was carried out with an Illumina HiSeq 2500 (Illumina, San Diego, CA, USA).

### 4.9. RNA-Seq and Enrichment Analysis of Differentially Expressed Genes

Raw data (raw reads) based on fastq format were initially processed with the use of in-house perl scripts. Clean data (clean reads) were procured by eliminating reads containing adapter and poly-N as well as reads of low quality from the raw data. The Q20, Q30, GC content, as well as level of sequence duplication of the clean data, were calculated. Analysis of downstream used clean data with high quality. Sequenced clean reads were mapped against predicted transcripts of the *A. flavus* NRRL 3357 genome1 (http://www.ncbi.nlm.nih.gov/genome/?term=aspergil-lus+flavus) using TopHat V2.1.1 and Bowtie v2.2.5 [40], and only unique matches were allowed. The FPKM (Fragments Per Kb of exon per Million reads) method was used to calculate and normalize the expression levels of the gene [41]. The genes expressed differentially were analyzed with the R edge R package V3.6.2 [35], and both a twofold change cut-off and an adjusted *p*-value of ≤0.05 were put in place as thresholds. Enrichment analysis of differential expression was carried out with the use of the GO-TermFinder v0.86 [42]. GO terms (including molecular function, cellular component, and biological process) and the Kyoto Encyclopedia of Genes and Genomes (KEGG) pathways were recognized as well enriched among genes expressed differentially when their *p*-values were ≤0.05.

### 4.10. Validation of RNA-Seq Analysis by Quantitative Real-Time PCR (qRT-PCR)

The totality of RNA was then separated by use of Trizol reagent (Invitrogen, Carlsbad, CA, USA). Briefly, the qRT-PCR conditions were thus: 95 °C for 10 min and 40 cycles of 95 °C for 15 s and 60 °C for 60 s. The fold or percentage of change in the relative expression of the mRNA of the target gene was assessed by the 2^−ΔΔCt^ approach. The gene-specific primers are listed in Appendix A.

### 4.11. Statistical Analysis

Data were expressed as mean ± standard deviation. Statistical analysis was carried with a one-way analysis of variance test for multiple comparisons. Differences between comparisons were deemed statistically significant at *p* < 0.05. SPSS software version 17.0 (SPSS Inc., Chicago, IL, USA) was deployed for analysis of data.

## Figures and Tables

**Figure 1 toxins-11-00154-f001:**
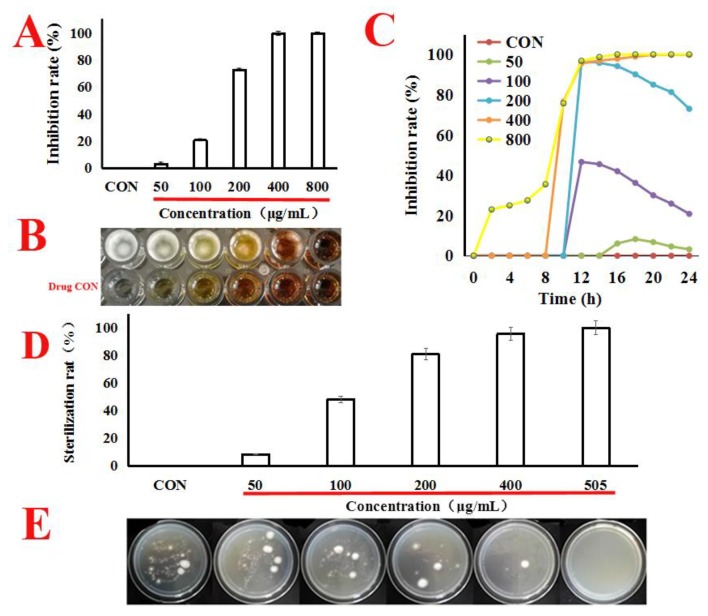
Quercetin inhibits the proliferation of *A. f lavus.* (**A**,**B**) *A. flavus* cells were treated with quercetin from 50 μg/mL to 800 μg/mL for 24 h at 30 °C. MIC value was calculated using SPSS 17.0. CON (untreated whih quercetin). (**C**) *A. flavus* cells were treated with quercetin from 50 μg/mL to 800 μg/mL for 24 h at 30 °C. For each treatment, the growth of *A. f lavus* was determined by automated absorbance measurements at 600 nm, detected absorption value every hour. (**D**,**E**) *A. flavus* cells were treated with quercetin (50, 100, 200, 400 and 505 μg/mL) for 24 h at 30 °C, the *A. flavus* solution was sucked out of the 96-well plate, centrifuged at 8000 rpm for 5 min, washed with quercetin, and then suspended the *A. flavus* with 0.9% normal saline. The washed *A. flavus* was then coated onto potato dextrose agar (PDA) plates, and the single colony was cultured for counting. All data were expressed as mean ± standard deviation (*n* = 3).

**Figure 2 toxins-11-00154-f002:**
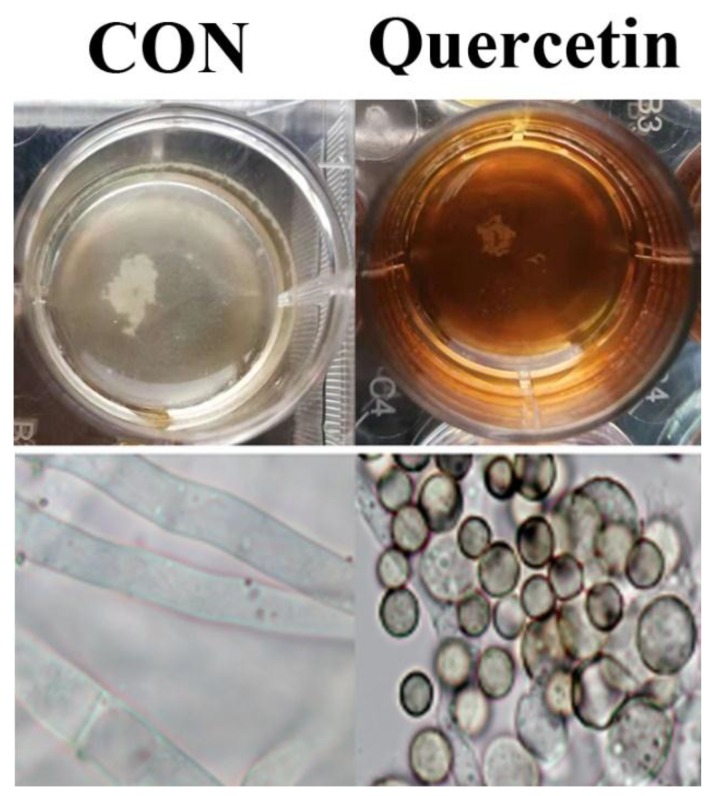
Morphological changes of *A. flavus*. *A. flavus* cells were treated with quercetin at 200 μg/mL for 24 h at 30 °C, and then the morphological changes of *A. flavus* were observed by the light microscope with a 100-fold oil mirror. CON (untreated with quercetin).

**Figure 3 toxins-11-00154-f003:**
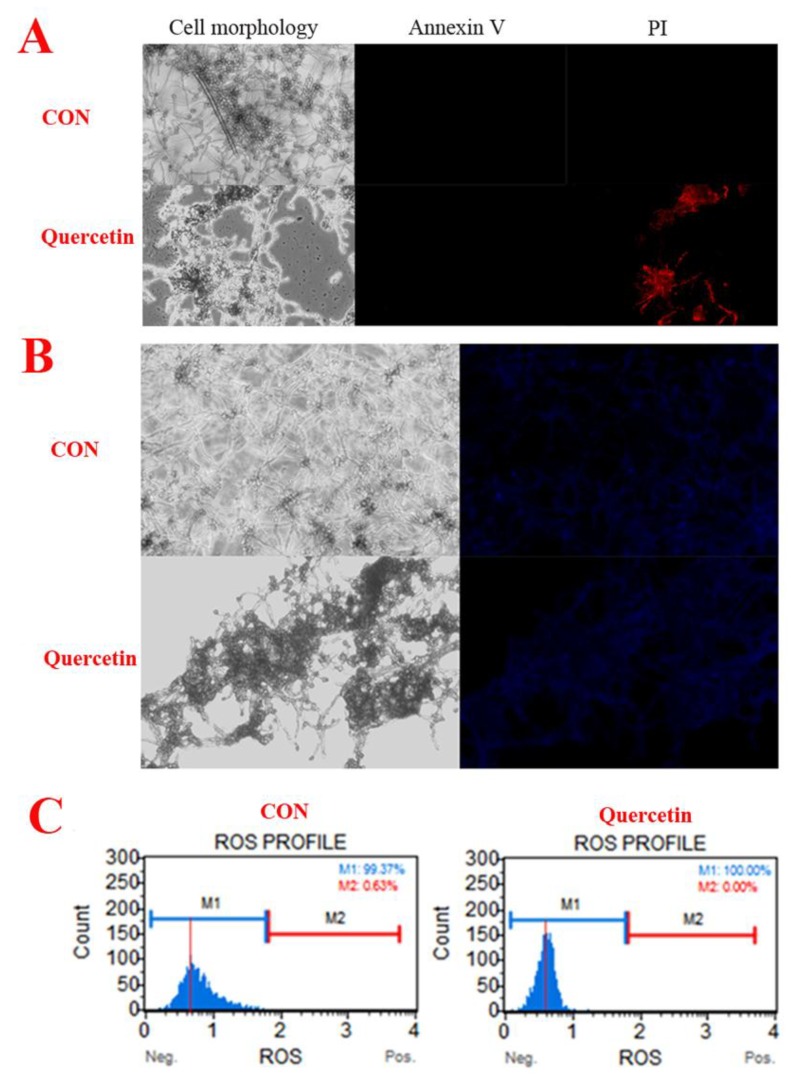
Quercetin induces *A. flavus* apoptosis. (**A**) Phosphatidylserine externalization, Spores (10^7^ CFU/mL) were treated with quercetin at 200 μg/mL. After 24 h, *A. flavus* cells were harvested and double-stained for 30 min with Annexin V-FITC/PI, to test for apoptosis. The cells were analyzed by a fluorescence microscope (20×). (**B**) Hochest 33342. Spores (10^7^ CFU/mL) were treated with quercetin for 24 h at 200 μg/mL, and then *A. flavus* cells were stained with Hochest 33342, a blue fluorescent dye to stain DNA, to test for nuclear. The cells were analyzed by a fluorescence microscope (20×). (**C**) Reactive oxygen species. Spores (10^7^ CFU/mL) were treated with quercetin for 24 h at 200 μg/mL, and then *A. flavus* cells were stained and analyzed by using Muse® Cell Analyzer. *A. flavus* cells untreated with quercetin were used as the control.

**Figure 4 toxins-11-00154-f004:**
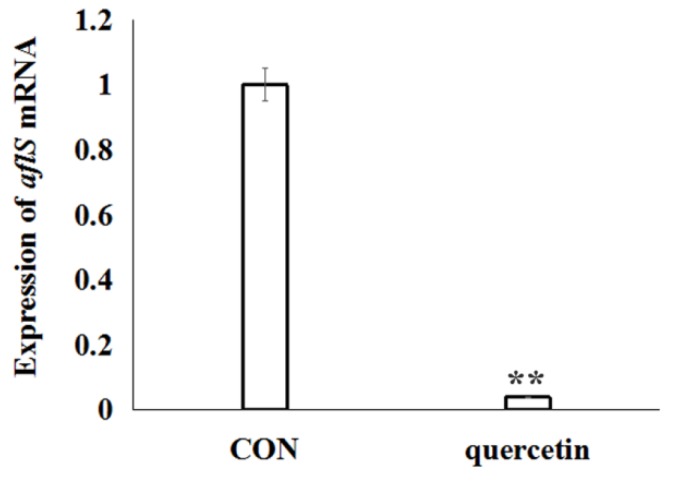
Relative lives of *aflS* mRNA from *A. flavus* exposed to quercetin for 24 h. The expression of *aflS* was quantified by SYBR quantitative polymerase chain reaction (qPCR) assay. *A. flavus* cells untreated with quercetin were used as the control. Data are presented with mean ± standard deviation (*n* = 5). ** *p* < 0.01, compared with the control group.

**Table 1 toxins-11-00154-t001:** Transcriptional activity of genes involved in *A. flavus* development.

Gene ID	Gene Length	Means-CK	Means-QT	log2(Fold Change)	Probability	Description
AFLA_020210	1362	370.1433333	64.65666667	−2.517213007	0.885380738	sexual development transcription factor NsdD
AFLA_048650	2100	156.1633333	38.95	−2.00336052	0.844491613	sexual development transcription factor SteA
AFLA_018540	1068	67.02666667	14.44666667	−2.213998536	0.81499661	G protein complex alpha subunit GpaB
AFLA_046990	2370	200.3033333	75.34333333	−1.410634662	0.8049291	APSES transcription factor StuA
AFLA_083110	255	766.0933333	24.60333333	−4.960594472	0.950709	conidiation-specific protein Con-10
AFLA_044790	309	525.2933333	124.2433333	−2.079954833	0.873234869	conidiation-specific family protein

CK, untreated with quercetin; QT, treated with quercetin.

**Table 2 toxins-11-00154-t002:** Ribosomal protein of differentially expressed genes when *A. flavus* was treated with quercetin.

NO.	Gene Name	Means-CK	Means-QT	log2(Fold Change)	Up-Down-Regulation (QT/CK)	Description
1	AFLA_047870	2358.77	2.84	−1.920818	Down	40S ribosomal protein S1
2	AFLA_034340	976.7166667	6.739345	−2.16111509	Down	40S ribosomal protein S10b
3	AFLA_050800	723.8966667	1.4697534	−1.69897	Down	40S ribosomal protein S12
4	AFLA_075030	802.26	9.62712	−1.92081875	Down	40S ribosomal protein S13
5	AFLA_044170	765.6133333	3.8280665	−2.30102999	Down	40S ribosomal protein S14
6	AFLA_044530	2185.433333	34.966928	−1.795880017	Down	40S ribosomal protein S15
7	AFLA_044110	763.1666667	9.9211671	−1.886056647	Down	40S ribosomal protein S16
8	AFLA_125890	2183.286667	43.66574	−1.69897	Down	40S ribosomal protein S17
9	AFLA_009000	1360.496667	34.012425	−1.602059991	Down	40S ribosomal protein S18
10	AFLA_050650	898.7666667	20.6716341	−1.638272164	Down	40S ribosomal protein S19
11	AFLA_117990	651.8966667	23.4682812	−1.443697016	Down	40S ribosomal protein S2
12	AFLA_043150	1318.843333	44.840662	−1.468521082	Down	40S ribosomal protein S20
13	AFLA_092120	8890.06	515.62348	−1.236572001	Down	40S ribosomal protein S21
14	AFLA_083740	680.39	6.8039	−2.067506415	Down	40S ribosomal protein S22
15	AFLA_021730	846.6866667	3.3867468	−2.399868277	Down	40S ribosomal protein S23
16	AFLA_071320	1901.14	7.60456	−2.397940008	Down	40S ribosomal protein S24
17	AFLA_083050	5390.12	29.64566	−2.259637311	Down	40S ribosomal protein S25
17	AFLA_127800	4272.09	26.059749	−2.214670164	Down	40S ribosomal protein S26
18	AFLA_083470	140.9033333	2.5362594	−1.744727494	Down	40S ribosomal protein S28
19	AFLA_101020	1328.043333	55.77806	−1.37675071	Down	40S ribosomal protein S4
20	AFLA_084620	1535.576667	16.89347	−1.958607314	Down	40S ribosomal protein S5
21	AFLA_026830	2783.25	125.24625	−1.346787486	Down	40S ribosomal protein S6
22	AFLA_029020	3209.673333	8954.783333	−1.48148606	Down	40S ribosomal protein S8
23	AFLA_101160	832.2533333	1.747731999	−2.677780753	Down	40S ribosomal protein S9
24	AFLA_030140	259.6833333	929.63	1.839903203	Down	60S acidic ribosomal protein P0
25	AFLA_127860	264.4766667	10.05011333	−1.420216403	Down	60S acidic ribosomal protein P1
26	AFLA_044520	984.8466667	2.659086	−2.568636235	Down	60S acidic ribosomal protein P2
27	AFLA_068000	765.0433333	1.606590999	−2.677780705	Down	60S ribosomal protein L1
28	AFLA_112090	1100.716667	7.1546583	−2.187086643	Down	60S ribosomal protein L11
29	AFLA_080140	1298.606667	37.659593	−1.537602002	Down	60S ribosomal protein L12
30	AFLA_115110	943.4	9.24532	−2.008773924	Down	60S ribosomal protein L13
31	AFLA_056250	879.56	22.86856	−1.585026652	Down	60S ribosomal protein L14
32	AFLA_029260	633.1733333	16.14591999	−1.593459819	Down	60S ribosomal protein L15, putative
33	AFLA_050000	1516.85	32.612275	−1.66756154	Down	60S ribosomal protein L16
34	AFLA_041990	1251.423333	4533.676667	1.857111595	Up	60S ribosomal protein L17
35	AFLA_047440	2671.72	35.266704	−1.879426068	Down	60S ribosomal protein L18
36	AFLA_046970	2952.823333	8912.88	1.593796639	Up	60S ribosomal protein L19
37	AFLA_048810	2272.563333	163.6245599	−1.136677139	Down	60S ribosomal protein L2
38	AFLA_029450	670.29	7.37319	−1.958607314	Down	60S ribosomal protein L20
39	AFLA_101150	2427.973333	41.2755466	−1.769551078	Down	60S ribosomal protein L21
40	AFLA_079880	233.58	1312.223333	2.490024624	Up	60S ribosomal protein L22
41	AFLA_092370	1086.093333	13.25033866	−1.913640169	Down	60S ribosomal protein L23
42	AFLA_092370	1387.563333	70.76572998	−1.292429823	Down	60S ribosomal protein L24
43	AFLA_048140	2383.7	165.66715	−1.155522824	Down	60S ribosomal protein L25
44	AFLA_110470	188.4966667	9.481382335	−1.298432014	Down	60S ribosomal protein L25, putative
45	AFLA_060150	2232.186667	13.39312	−2.221848749	Down	60S ribosomal protein L26
46	AFLA_127220	862.0833333	13.44849999	−1.806875401	Down	60S ribosomal protein L27
47	AFLA_054760	1163.723333	8.0296909	−2.161150909	Down	60S ribosomal protein L28
48	AFLA_103770	1127.98	10.039022	−2.050609993	Down	60S ribosomal protein L28
49	AFLA_134740	1447.586667	50.66553335	−1.455931955	Down	60S ribosomal protein L3
50	AFLA_045790	2011.47	43.246605	−1.66756154	Down	60S ribosomal protein L30
51	AFLA_045790	1371.076667	4046.166667	−1.561246503	Down	60S ribosomal protein L31
52	AFLA_003480	11,949.94667	33,519.81333	1.488009936	Up	60S ribosomal protein L32
53	AFLA_086630	1485.253333	2.673455999	−2.744727494	Down	60S ribosomal protein L33
54	AFLA_086630	2061.426667	18.55284	−2.04575749	Down	60S ribosomal protein L34
55	AFLA_086630	1270.083333	5.715374999	−2.346787486	Down	60S ribosomal protein L35
56	AFLA_112390	832.9033333	4.164516667	−2.301029995	Down	60S ribosomal protein L36
57	AFLA_112390	3332.77	39.99324	−1.920818753	Down	60S ribosomal protein L37
58	AFLA_112390	1112.88	14.244864	−1.89279003	Down	60S ribosomal protein L38
59	AFLA_112390	1971.986667	29.77699867	−1.821023052	Down	60S ribosomal protein L42
60	AFLA_112390	3816.61	106.483419	−1.554395796	Down	60S ribosomal protein L43
61	AFLA_018700	1022.05	17.885875	−1.732828271	Down	60S ribosomal protein L5
62	AFLA_068420	3303.43	87.540895	−1.576754126	Down	60S ribosomal protein L6
63	AFLA_041710	1208.193333	5.255640998	−2.361510743	Down	60S ribosomal protein L7
64	AFLA_033980	1874.96	88.12312	−1.327902142	Down	60S ribosomal protein L8
65	AFLA_088370	686.0866667	3323.653333	2.276307178	Up	60S ribosomal protein L9

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
