# Peer review of "Quercetin Inhibits the Proliferation and Aflatoxins Biosynthesis of Aspergillus flavus"

_toxins, 2019, doi:10.3390/toxins11030154_

Round 1
Reviewer 1 Report
Title: Quercetin Inhibits the Proliferation and Aflatoxins 2 Biosynthesis of Aspergillus flavus
Recommendation: Accept after major revisions
The current manuscript titled “Quercetin Inhibits the Proliferation and Aflatoxins Biosynthesis of Aspergillus flavus” studies the effect of quercetin’s anti-proliferation action in A. flavus. The manuscript describes how quercetin can effectively hamper the proliferation of A. flavus in dose-effect and time-effect relationships. This is an important field of study as these results provide some theoretical basis for the use of quercetin as an anti-mildew agent resource. The results described in the manuscript are significant. However, the outline of the manuscript needs some revision before it can accepted in its current form by the journal.
Major comments:
· The manuscript needs to have a separate discussion and conclusion section. In the current form the authors have merged the discussion section with the individual results section. A separate section will help to compare this study with other studies in the same field. Additionally, it will highlight the significance of the results outlined in this study.
· The figure legends on figure 1A and 1B needs to be revised to “inhibition Rate”.
· Few key references are missing for the manuscript discussion section. The references need to added when a separate discussion section is added in the manuscript.
· Each figure legend need to outline how the statistical analysis was done.
Author Response
Dear reviewer,
Thank you for your useful comments and suggestions on our manuscript. We have modified the manuscript accordingly, and the detailed corrections are listed below point by point:
1. The manuscript needs to have a separate discussion and conclusion section. In the current form the authors have merged the discussion section with the individual results section. A separate section will help to compare this study with other studies in the same field. Additionally, it will highlight the significance of the results outlined in this study.
The authors’ Answer: Thank you for your great suggestions, your suggestions are of great help to us, we have separated result and discussion section in the revised manuscript.
2. The figure legends on figure 1A and 1B needs to be revised to “inhibition Rate”.
The authors’ Answer: We are sorry for our neglectfulness, the figure legends on figure 1A and 1B have been amended in revised manuscript.
3. Few key references are missing for the manuscript discussion section. The references need to added when a separate discussion section is added in the manuscript.
The authors’ Answer: Thank you for your great suggestions, your suggestions are of great help to us, we have added few key references for the manuscript discussion section in the revised manuscript.
4. Each figure legend need to outline how the statistical analysis was done.
The authors’ Answer: Thank you for your great suggestions, your suggestions are of great help to us, we have added the statistical analysis in each figure legends of the revised manuscript.
Reviewer 2 Report
Authors are kindly asked to improve figure captions in order to figures can be self-explanatory.
Please, recheck the copyright issue and citation for figure used from KEGG.
Please, try to connect the obtained results with the available literature reports, and in some instances explain results a bit more.
Please, improve part of materials and methods (3.7)
All, comments and suggestions are marked within the revised document.

Author Response
Dear reviewer,
Thank you for your useful comments and suggestions on our manuscript. We have modified the manuscript accordingly, and the detailed corrections are listed below point by point:
1. Authors are kindly asked to improve figure captions in order to figures can be self-explanatory. Please, recheck the copyright issue and citation for figure used from KEGG.
The authors’ Answer: Thank you for your great suggestions, your suggestions are of great help to us, we have improved figure captions in the revised manuscript. In addition, to avoid copyright issue, we have presented the data in table form in the revised manuscript.
2. Please, try to connect the obtained results with the available literature reports, and in some instances explain results a bit more
The authors’ Answer: Thank you for your great suggestions, your suggestions are of great help to us, we have tried to explain the obtained results with the available literature reports in the revised manuscript.
3. Please, improve part of materials and methods (3.7)
The authors’ Answer: Thank you for your great suggestions, your suggestions are of great help to us, we have improved part of materials and methods (3.7) in the revised manuscript.
Round 2
Reviewer 2 Report
Authors made correction as suggested.